# Disentangling behavioral determinants of seasonal influenza vaccination in post-corona era: An integrated model approach

So-Hyun Kim[1], Minsoo Jung [1,2]*

1 Department of Health Science, Dongduk Women's University, Seoul, South Korea, 2 Center for Community-Based Research, Dana-Farber Cancer Institute, Boston, Massachusetts

* mins.jung@gmail.com, mj748@dongduk.ac.kr

## Abstract

Seasonal influenza vaccination (SIV) is influenced by various factors, including socio-demographic characteristics and socioeconomic status of the recipient. Nevertheless, *in the post-COVID-19 era, the importance of vaccination and group immunity has grown*. Therefore, applying an integrated model to identify behavioral determinants of vaccination is needed. This study aimed to identify contextual factors affecting SIV by applying Andersen's model. We utilized secondary national datasets (n = 14,535) from the 2022 Community Health Survey conducted by the Korea Disease Control and Prevention Agency. Predisposing factors were gender and age. Enabling factors were income, educational attainment, and marital status. Need factors were presence of chronic disease, health risk behaviors (smoking and/or drinking alcohol), physical activity, and coronavirus disease 2019 (COVID-19) vaccination status. Dependent variable was influenza vaccination status. Multiple binomial logistic regression analyses were performed to identify predictors of influenza vaccination status among Korean adults, stratified by gender and age. According to the results, in men, higher education increased the likelihood of influenza vaccination by 1.089 times (95% CI: 1.000–1.185), while being married increased it by 1.619 times (95% CI: 1.413–1.856); however, smoking and binge drinking reduced the likelihood by 0.822 times (95% CI: 0.732–0.923) and 0.749 times (95% CI: 0.650–0.864), respectively. Among young men, marriage (OR=1.480, 95% CI: 1.131–1.935) and physical activity (OR=1.381, 95% CI: 1.053–1.811) were significant positive factors, while among older men, chronic disease presence increased vaccination likelihood by 1.339 times (95% CI: 1.126–1.592). In women, higher education (OR=1.168, 95% CI: 1.075–1.270) and marriage (OR=2.242, 95% CI: 1.965–2.557) were strong positive predictors, while COVID-19 vaccination history consistently increased influenza vaccination likelihood (OR=1.852, 95% CI: 1.712–2.003). Among young women, smoking reduced vaccination likelihood (OR=0.551, 95% CI: 0.359–0.847), while among older women, having a chronic disease increased vaccination likelihood by 1.354

**Data availability statement:** "Relevant data are within the paper and its Supporting Information files. The raw data used in this study are from the Community Health Survey and can be downloaded from the Korea Disease Control and Prevention Agency (KDCA) website (https://chs.kdca.go.kr/chs/main.do). Researchers submit a data use plan to KDCA for review, after which access to the data is granted via an individual download link within 2–3 days."

**Funding:** The author(s) received no specific funding for this work.

**Competing interests:** The authors have declared that no competing interests exist.

times (95% CI: 1.133–1.619). This study empirically reveals that SIV is affected by predisposing, enabling, and need factors. To effectively intervene in individual health behaviors, it is necessary to identify characteristics of the population, provide segmented messages, and apply customized strategies.

## Introduction

Seasonal influenza, commonly referred to as the flu, is a highly contagious respiratory illness caused by influenza viruses. Influenza remains a major public health threat despite the availability of vaccines and antiviral medications, causing significant morbidity and mortality, especially among vulnerable populations. The basic reproduction number (R0) of seasonal influenza typically ranges from 1.3 to 1.8, meaning that each infected person can, on average, infect more than one other person, leading to rapid community spread [1]. The potential for influenza viruses to mutate and reassort that can lead to the emergence of new strains poses a continuous threat of pandemics, as seen in the 2009 H1N1 pandemic. Certain groups, including the elderly, young children, pregnant women, and individuals with chronic health conditions (e.g., asthma, diabetes, heart disease), are at higher risk for severe influenza-related complications, including pneumonia, exacerbation of existing chronic conditions, and even death. For instance, the Centers for Disease Control and Prevention (CDC) estimated that during the 2019–2020 flu season in the United States, there were approximately 38 million flu illnesses, 400,000 hospitalizations, and 22,000 deaths, with a significant proportion occurring in the elderly population [2].

Seasonal influenza exerts a substantial burden on healthcare systems. During peak flu season, hospitals and clinics often experience increased admissions due to influenza-related complications. This can strain resources, particularly during concurrent public health crises such as the coronavirus disease 2019 (COVID-19) pandemic. In severe flu seasons, increased demand for healthcare services can lead to overwhelmed emergency departments, shortages of intensive care unit beds, and increased mortality rates due to the inability to provide timely care to all patients [3]. Although vaccination is the primary preventive measure against influenza, effectiveness of flu vaccine can vary from year to year due to antigenic drift and shift in circulating strains. Vaccine effectiveness typically ranges from 40% to 60%, depending on the match between the vaccine and circulating strains [2]. Despite these limitations, vaccination remains a critical public health tool as it can reduce the severity of illness, prevent hospitalizations, and save lives. A meta-analysis of vaccine effectiveness from 2004 to 2015 has shown that influenza vaccines can reduce the risk of influenza-associated medical visits by an average of 41% overall [4]. A study published in Clinical Infectious Diseases has found that vaccination can reduce the risk of severe outcomes among hospitalized flu patients by 37% [5].

However, seasonal influenza vaccination coverage rates (VCRs) in major countries are often below 50%, much lower than the World Health Organization's (WHO) target of 75% [6]. It is reasonable to focus policy capacity on seasonal influenza

vaccination (SIV) on high-risk groups, including the elderly (age 65 or older), children, pregnant women, and people with chronic diseases. However, in the post-corona era, a twindemic in which both corona and seasonal influenza are prevalent at the same time can occur at any time. Thus, measures for protecting low-risk groups are also needed. SIV varies greatly across developed countries due to differences in healthcare systems, public health policies, vaccine availability, and cultural attitudes toward vaccination. The Korean government has increased vaccine accessibility to high-risk groups such as the elderly, pregnant women, and people with chronic diseases through the National Immunization Program, which provides vaccines free of charge [7]. However, no special policies or campaigns have been implemented for general adults. The influenza vaccination season in Korea generally begins in October. High-risk groups receive vaccinations at public health centers, while general adults receive vaccinations at clinics or hospitals. The cost generally ranges from 20,000 KRW to 40,000 KRW (approximately 15 USD to 30 USD) depending on the clinic and type of vaccine used. A quadrivalent influenza vaccine is usually used [8]. As of 2020, the VCR in Korea has reached 65–70% for the elderly (age 65 or older). However, the VCR is low at 25–35% for general adults aged 19–64 years.

SIV is influenced by a subject's gender, age, household composition, education level, income level, chronic disease, health status, and attitude and knowledge about vaccines [9–16]. However, systematic analysis of factors determining SIV has been lacking. Vaccination involves time, costs, and potential side effects, and individuals may still contract influenza despite being vaccinated. In addition, if one does not get vaccinated, he or she might be lucky without getting infected [17,18]. Therefore, decision-making models for health-promoting behaviors such as vaccination tend to be complex [15,18].

Nevertheless, in the post-Corona era, the value of vaccination and the need for group immunity have increased. Therefore, applying an integrated model to identify behavioral determinants of vaccination is needed. Andersen's behavioral model of health service utilization is an integrated model that can analyze factors used in medical services [19]. The model identifies three primary categories of factors that influence health services utilization: Predisposing factors – demographic characteristics such as age, gender, and social structure (e.g., education, occupation) that shape an individual's likelihood of seeking healthcare. Enabling factors – personal, family, and community resources that facilitate or hinder access to healthcare, including income, health insurance coverage, and healthcare facility availability. Need factors – both perceived and evaluated needs for medical care, encompassing symptoms, illness severity, and professional assessments by healthcare providers. Over time, Andersen's model has been expanded to include environmental factors and health behaviors, recognizing the complex interplay between individuals and their broader social and physical environments. The model can be used to analyze patterns of health services use across different populations and to identify barriers to access, with an aim to inform policies and interventions that promote equitable access to healthcare. Andersen's model has been applied to various disease prevention and health promotion behaviors [20–22].

The objective of this study is to analyze the factors influencing seasonal influenza vaccination rates among the general adult population in South Korea using Andersen's behavioral model of health service utilization. While previous research has identified demographic and socioeconomic factors affecting vaccination, systematic analysis of these determinants remains limited. By categorizing factors into predisposing characteristics, enabling resources, and need factors, this study seeks to identify disparities in vaccination coverage and explore the underlying behavioral and structural barriers. The findings aim to inform targeted interventions and policy recommendations to improve vaccination rates, particularly among low-coverage groups, in the post-COVID-19 era.

## Methods

### Datasets

We utilized secondary national datasets from the 2022 Community Health Survey conducted by the Korea Disease Control and Prevention Agency (formerly the Korea Centers for Disease Control and Prevention). Samples were collected using probability proportional cluster sampling. The sample size comprised 900 individuals from each of the 258 public

health centers nationwide, yielding a ± 3% desired sampling error with a 95% confidence level. At each sampling point, an average of five households were selected. All household members aged 19 years or older were interviewed. Stratification of the surveyed population was twofold: 1) by *dong/eup/myeon* (small administrative units) within jurisdictions of the 253 health centers located across the country; and 2) by housing units (apartments and houses). The sample design aimed to produce accurate statistics from a small-scale sample survey by leveraging stratification of the surveyed population [23]. Consequently, the Community Health Survey used a multistage, stratified, random sampling method to represent the broader Korean population [24]. In this stratification, *dong/eup/myeon* units within the jurisdictions of the 253 health centers nationwide were designated as primary sampling units (tiers) and housing types (apartments and general houses) were designated as secondary sampling units. The 2022 Community Health Survey was conducted from August 16, 2022 to October 31, 2022. Surveyors visited households in person and received survey responses through 1:1 interviews for a total of 138 questions.

The total number of respondents to the 2022 Community Health Survey was 231785. Initially, we excluded 208842 individuals residing outside of Seoul and 6552 individuals aged 65 years or more from the sample. Subsequently, among the 16391 adults living in Seoul, 396 individuals who declined to answer questions about influenza vaccination, income level, and education level were also excluded. Additionally, 1458 individuals who refused to provide information regarding their marital status or who reported being divorced, separated, or widowed were excluded. This exclusion was necessary because the study used age and gender as key stratification variables. In addition, it is uncommon for men and women in their 20s to experience divorce, separation, or bereavement. Finally, two cases that did not provide responses regarding physical activity and COVID-19 vaccination were excluded. Thus, the final sample used for analysis comprised 14535 individuals. The selection process for study subjects is summarized in a STROBE chart as shown in Fig 1 (Fig 1).

## Analytical framework

We explored behavioral determinants of influenza vaccination with Andersen's behavioral model of health services use. Andersen's behavioral model conceptualizes factors influencing access to medical care [19]. It is structured around three core components: predisposing characteristics, enabling resources, and need factors. Predisposing characteristics include demographics, social structure, and health beliefs. Enabling resources refer to personal, family, and community resources that facilitate or hinder access to healthcare. Need factors are an individual's perceived and evaluated health status that drives the utilization of health services. The model suggests that these components can interact with each other to determine health service usage and ultimately influence health outcomes.

Gender and age as predisposing factors, income, educational attainment, and marital status as enabling factors, and presence of chronic disease, health risk behaviors (smoking and/or drinking alcohol), physical activity, and COVID-19 vaccination status as need factors were analyzed in the present study based on previous studies [10,25,26]. The analytical framework of this study is shown in Fig 2 (Fig 2).

## Measures

This section outlined the measurement tools used in this study, including the dependent and independent variables.

**Dependent variables.** The dependent variable was influenza vaccination status. Respondents answered "yes" (1) or "no" (0) to the following question: "Have you ever been vaccinated against influenza (flu) in the past year?".

**Independent variables.** Based on the Andersen model as described above, independent variables were predisposing factors (gender, age), enabling factors (income, educational attainment, marital status), and need factors (presence of chronic disease, smoking, binge drinking, physical activity, COVID-19 vaccination).

Of the two predisposing factors, gender was divided into 'men' and 'women' and age was classified into 'young adult' (19–34 years old), 'middle-aged adult' (35–49 years old), and 'older adult' (50–64 years old) based on standards for each life cycle [27].

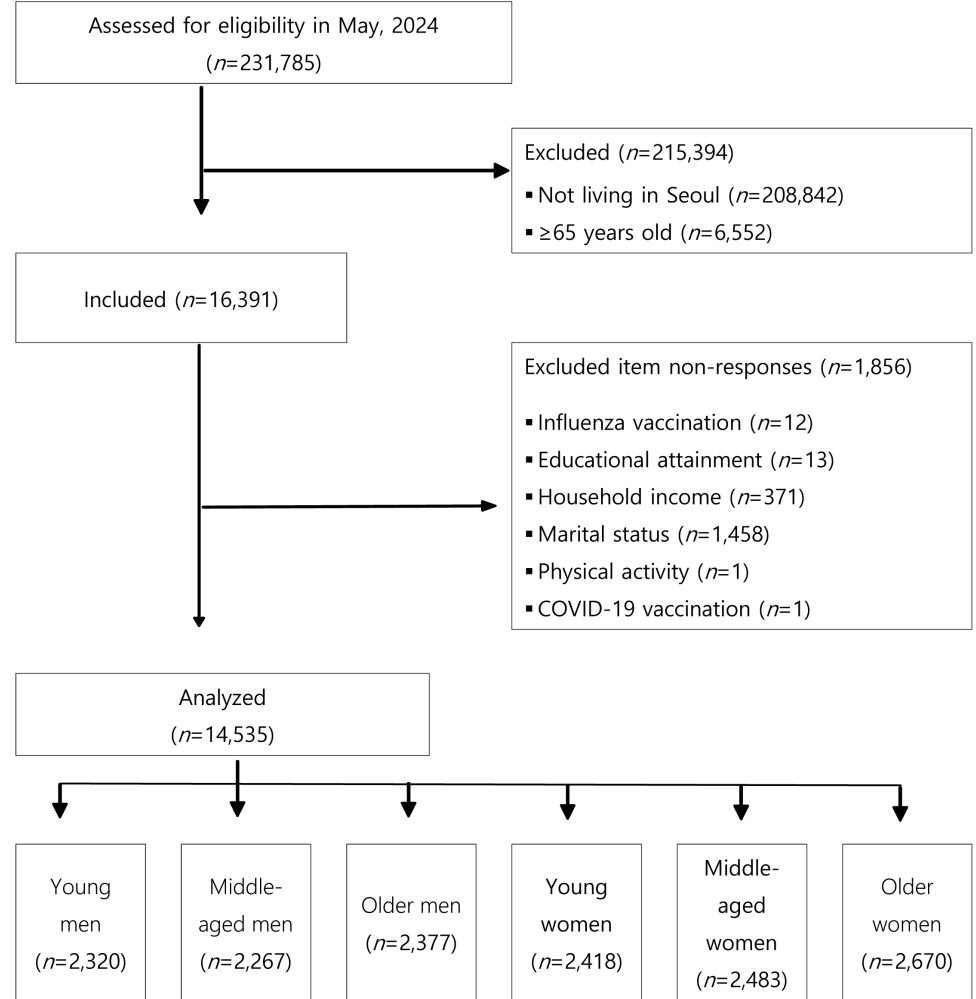

**Fig 1. STROBE flow chart of this study.**

For enabling factors, income level was assessed by asking an open-ended question: "Including all income, wages, real estate, pension, interest, government subsidies, and pocket money from relatives or children, what is approximately the total income of your household in the past year?". Since this was a household's average monthly income, we adjusted the income amount by the number of respondents' household members and reclassified the bottom 25% as 'low income' (1), the middle 50% as 'middle income' (2), and the top 25% as 'high income' (3). Educational attainment was measured with the following two questions: "How long have you been attending school?" and "Have you graduated from that school?" Those who answered 'elementary school or less', 'middle school', or 'high school' were classified as 'high school graduate or less' (1). Those who answered '2/3-year college' or '4-year college' were classified as 'college graduate' (2). Those who answered 'graduate school attendance, completion, graduation' were classified as 'graduate school or higher' (3). Marital status was measured by answering the following question, "Which of the following does your current marital status correspond to?" as "Has a spouse and living together (including common-law marriage)" or "Unmarried and has no spouse." The former was classified as 'married' (1) and the latter was classified as 'single' (0).

In the case of need factors, a respondent's presence of chronic disease was measured with the following two questions: "Have you ever been diagnosed with high blood pressure by a doctor?" and "Have you ever been diagnosed with

**Fig 2. Analytical framework of this study.**

diabetes by a doctor?" Those who did not have hypertension or diabetes were classified as 'none' (0) and those who had been diagnosed with either disease were classified as 'present' (1).

A respondent's smoking status was measured with the following two questions: "How many cigarettes have you smoked in your entire life?" and "Do you currently smoke cigarettes?" Among those who responded that the total amount of cigarettes they had ever smoked was 'more than 5 packs (100 cigarettes)', those who responded that they were currently smoking cigarettes 'every day' or 'sometimes' were classified as 'smokers' (1). Otherwise, they were classified as 'non-smokers' (0). Respondents' binge drinking was measured with the following three questions: "Have you ever had more than one drink in your life?", "How often do you drink alcohol?", and "How much alcohol do you drink at one time?". Binge drinking is defined as drinking more than seven glasses of soju at a time for men and more than five glasses of soju at a time for women more than twice a week [28]. Accordingly, if the respondent was a binge drinker, it was coded as 1. Otherwise, it was coded as 0.

Physical activity was divided into vigorous physical activity, moderate physical activity, and walking. Vigorous physical activity was measured with the following two questions: "During the past week, on how many days did you do 10 minutes or more of vigorous physical activity that made you feel very tired or out of breath more than usual?" and "How many minutes do you usually do this kind of vigorous physical activity per day?" Moderate physical activity was measured with the following two questions: "During the past week, on how many days did you do 10 minutes or more of vigorous physical activity that made you feel a little more tired or out of breath than usual (excluding walking)?" "How many minutes of moderate physical activity do you do on a typical day?" Walking was measured by two questions: "During the past week, on how many days did you walk at least 10 minutes at a time?" and "How long do you usually walk on one of these days?" When including amounts of all three physical activities described above, we considered practicing physical activity as 'yes' (1) if the total amount of physical activity per week was 600 MET-min-wk-1 or more. Otherwise, it was coded as 'no' (0). Based on the International Physical Activity Questionnaire, we calculated the total amount of physical activities by multiplying the MET level of each activity, the number of minutes of exercise, and the number of times per week [29].

MET stands for metabolic equivalent. It indicates the intensity of physical activity based on the amount of oxygen required during physical activity. Based on previous research, MET was set at 8.0 for vigorous physical activity, 4.0 for moderate physical activity, and 3.3 for walking physical activity [29].

Respondents' COVID-19 vaccination status was measured with the following question: "Have you ever been vaccinated against COVID-19?" Response categories were as follows: "Yes, I received the 1st dose", "Yes, I received the 2nd dose (including Janssen)", "Yes, I received the 3rd dose (booster shot)", "Yes, I received the 4th dose", and "No, I have not been vaccinated". We classified responses into '1st vaccination or less', '2nd vaccination', '3rd vaccination', and '4th vaccination or more' according to survey question guidelines of the 2022 Community Health Survey. Those with '1st vaccination or less' (1) refers to those who have not been vaccinated and those who have received only one dose of Astra-Zeneca, Pfizer, Moderna, or Novavax. Those with '2nd vaccination' (2) included those who received Janssen as the first vaccination and those who received one vaccine from AstraZeneca, Pfizer, Moderna, or Novavax as the first and second vaccinations. Those with the '3rd vaccination' (3) included people who received Janssen as the first vaccination and one vaccine from AstraZeneca, Pfizer, Moderna, or Novavax as the second vaccination, as well as those who received a booster shot. Those with the '4th vaccination or more' (4) included those who had received one vaccine from Pfizer, Moderna, or Novavax among those with 3rd vaccinations.

### Statistical analyses

Initially, descriptive statistical analyses were performed to examine social and behavioral characteristics of participants. Concurrently, cross tabulation and bivariate analyses were conducted to assess associations between respondents' predisposing, enabling, and need factors and their SIV status. Subsequently, a chi-square test was utilized to investigate the association between gender, age, and influenza vaccination status within the sample. Finally, multiple binomial logistic regression analyses were performed to identify predictors of influenza vaccination status among Korean adults stratified by gender and age. All statistical analyses were executed using STATA v. 14.0 (STATA, College Station, TX, USA).

### Ethics statement

The community health survey raw data is secondary data disclosed to the public and is not subject to human subject research based on Article 2, Paragraph 2 of the Enforcement Rules of the Bioethics and Safety Act, and is therefore excluded from IRB (Institutional Review Board) review. The original IRB approval number from the Korea Disease Control and Prevention Agency was 2016-10-01-P-A. We also received approval for review exemption from the Dongduk Women's University Institutional Review Board (DDWU2408-01).

## Results

### General characteristics of the sample

General characteristics of the respondents are shown in Table 1 (Table 1). Regarding the gender, 'men' and 'women' accounted for 47.9% and 52.1% of respondents, respectively. Regarding the age of respondents, 'young adults (19-34 years old)', 'middle-aged adults (35-49 years old)', and 'older adults (50-64 years old)' accounted for 32.6%, 32.7%, and 34.7%, respectively. Regarding the income level of respondents, those with 'lower income', 'middle income', and 'upper income' accounted for 25.1%, 49.9%, and 25.0%, respectively. Regarding the educational attainment of respondents, 37.8%, 52.9%, and 9.3% were 'high school graduate or less', 'college graduate', and 'graduate school or higher', respectively. In terms of marital status, 39.3% were 'single' and 60.7% were 'married'. Regarding the presence or absence of chronic diseases, 83.8% of respondents said 'no' and 16.2% said 'yes'. Regarding the current smoking status of respondents, 83.8% said 'no' and 16.2% said 'yes'. Regarding binge drinking, 89.3% said 'no' and 10.7% said 'yes'. Regarding whether respondents practiced physical activity, 22.2% said 'no' and 77.8% said 'yes'. Regarding COVID-19 vaccination

**Table 1. General characteristics of the sample (n = 14,535).**

|  | Categories |  | n | % |
|---|---|---|---|---|
| Predisposing factors | Gender | Men | 6,964 | 47.9 |
|  |  | Women | 7,571 | 52.1 |
|  | Age | Young adult (19–34 yrs) | 4,738 | 32.6 |
|  |  | Middle-aged adult (35–49 yrs) | 4,750 | 32.7 |
|  |  | Older adult (50–64 yrs) | 5,047 | 34.7 |
| Enabling factors | Income | Lower income (bottom 24%) | 3,643 | 25.1 |
|  |  | Middle income | 7,260 | 49.9 |
|  |  | Upper income (top 25%) | 3,632 | 25.0 |
|  | Educational attainment | High school graduates or less | 5,491 | 37.8 |
|  |  | College graduates | 7,690 | 52.9 |
|  |  | Graduate school or higher | 1,354 | 9.3 |
|  | Marital status | Single | 5,718 | 39.3 |
|  |  | Married | 8,817 | 60.7 |
| Need factors | Chronic disease | No | 12,181 | 83.8 |
|  |  | Yes | 2,354 | 16.2 |
|  | Smoking | No | 12,186 | 83.8 |
|  |  | Yes | 2,349 | 16.2 |
|  | Binge drinking | No | 12,986 | 89.3 |
|  |  | Yes | 1,549 | 10.7 |
|  | Physical activity | No | 3,222 | 22.2 |
|  |  | Yes | 11,313 | 77.8 |
|  | COVID-19 vaccination[+] | 1st vaccination or less | 701 | 4.8 |
|  |  | 2nd vaccination | 4,006 | 27.6 |
|  |  | 3rd vaccination | 8,740 | 60.1 |
|  |  | 4th vaccination or more | 1,088 | 7.5 |
| Seasonal influenza vaccination |  | No | 8,775 | 60.4 |
|  |  | Yes | 5,760 | 39.6 |
| Total |  |  | 14,535 | 100.0 |

[+]1st vaccination or less: Those who have not been vaccinated and those who have received a total of 1 dose of AstraZeneca, Pfizer, Moderna, or Novavax

2nd vaccination: Those who received the first dose of Janssen and those who received a total of 2 doses of AstraZeneca, Pfizer, Moderna, and Novavax

3rd vaccination: Among those vaccinated with the first dose of Janssen, those vaccinated with AstraZeneca, Pfizer, Moderna, Novavax and those vaccinated with a booster shot

4th vaccination or higher: Among those receiving the 3rd vaccination, those vaccinated with Pfizer, Moderna, or Novavax

status of respondents, 4.8%, 27.6%, 60.1%, and 7.5% had '1st vaccination or less', '2nd vaccination', '3rd vaccination', and '4th vaccination or more', respectively. Regarding whether respondents had been vaccinated against seasonal influenza, 60.4% said 'no' and 39.6% said 'yes', meaning that more people had not been vaccinated against seasonal influenza.

## Associations of predisposing, enabling, and need factors with influenza vaccination behavior

Results of a chi-square test to analyze relationships of respondents' predisposing factors, enabling factors, and need factors with influenza vaccination behavior are shown in Table 2 (Table 2). Looking at predisposing factors, women

**Table 2. Association between predisposing, enabling, and need factors and influenza vaccination behavior (n = 14,535).**

| Categories | | | Unvaccinated | Vaccinated | χ² |
|---|---|---|---|---|---|
| | | | n (%) | n (%) | |
| Predisposing factors | Gender | Men | 4,501(64.6) | 2,463(35.4) | 101.456*** |
| | | Women | 4,274(56.5) | 3,297(43.5) | |
| | Age | Young adult (19–34) | 3,368(71.1) | 1,370(28.9) | 340.917*** |
| | | Middle-aged (35–49) | 2,668(56.2) | 2,082(43.8) | |
| | | Older adult (50–64) | 2,739(54.3) | 2,308(45.7) | |
| Enabling factors | Income | Lower income | 2,246(61.7) | 1,397(38.3) | 14.146*** |
| | | Middle income | 4,431(61.0) | 2,829(39.0) | |
| | | Upper income | 2,098(57.8) | 1,534(42.2) | |
| | Educational attainment | High school graduate or less | 3,353(61.1) | 2,138(38.9) | 31.005*** |
| | | College graduate | 4,700(61.1) | 2,990(38.9) | |
| | | Graduate school or higher | 722(53.3) | 632(46.7) | |
| | Marital status | Single | 4,073(71.2) | 1,645(28.8) | 464.658*** |
| | | Married | 4,702(53.3) | 4,115(46.7) | |
| Need factors | Chronic disease | No | 7,567(62.1) | 4,614(37.9) | 96.258*** |
| | | Yes | 1,208(51.3) | 1,146(48.7) | |
| | Smoking | No | 7,154(58.7) | 5,032(41.3) | 87.353*** |
| | | Yes | 1,621(69.0) | 728(31.0) | |
| | Binge drinking | No | 7,723(59.5) | 5,263(40.5) | 41.235*** |
| | | Yes | 1,052(67.9) | 497(32.1) | |
| | Physical activity | No | 1,966(61.0) | 1,256(39.0) | n.s |
| | | Yes | 6,809(60.2) | 4,504(39.8) | |
| | COVID-19 vaccination+ | 1st vaccination or less | 553(78.9) | 148(21.1) | 646.536*** |
| | | 2nd vaccination | 2,861(71.4) | 1,145(28.6) | |
| | | 3rd vaccination | 4,984(57.0) | 3,756(43.0) | |
| | | 4th vaccination or more | 377(34.7) | 711(65.3) | |

1) *: p < 0.05, **: p < 0.01, ***: p < 0.001

2) n.s: not significant at p < 0.05

+1st vaccination or less: Those who have not been vaccinated and those who have received a total of 1 dose of AstraZeneca, Pfizer, Moderna, or Novavax

2nd vaccination: Those who received the first dose of Janssen and those who received a total of 2 doses of AstraZeneca, Pfizer, Moderna, and Novavax

3rd vaccination: Among those vaccinated with the first dose of Janssen, those vaccinated with AstraZeneca, Pfizer, Moderna, Novavax and those vaccinated with a booster shot

4th vaccination or higher: Among those receiving the 3rd vaccination, those vaccinated with Pfizer, Moderna, or Novavax

were 8.1%p more likely to have received influenza vaccination than men and older adults were 16.8%p more likely to have received influenza vaccination than young adults. Looking at possible factors, the upper income group was 3.9%p more likely to receive influenza vaccination than the lower income group and the group with a graduate school education or higher was 7.8%p more likely to receive influenza vaccination than the group with a high school education or lower. Additionally, the vaccination rate in the married group was 17.9%p higher than in the single group. Looking at need factors, the vaccination rate in the group with chronic diseases was 10.8%p higher than in the healthy group. The non-smoker group had a 10.3%p higher vaccination rate than the smoker group. The non-binge drinker group had a 8.4%p higher vaccination rate than the binge drinker group. In addition, the group with the 4th or higher level of COVID-19 vaccination received 44.2%p more influenza vaccination than the group with the 1st vaccination of COVID-19 or lower.

**Table 3. Association between gender and age and influenza vaccination behavior (n = 14,535).**

| Categories | | Unvaccinated | Vaccinated | $\chi^2$ |
|---|---|---|---|---|
| | | n (%) | n (%) | |
| Men | Young adult (19–34 years old) | 1,696(73.1) | 624(26.9) | 114.023*** |
| | Middle-aged adult (35–49) | 1,405(62.0) | 862(38.0) | |
| | Older adult (50–64) | 1,400(58.9) | 977(41.1) | |
| Women | Young adult (19–34 years old) | 1,672(69.1) | 746(30.9) | 233.197*** |
| | Middle-aged adult (35–49) | 1,263(50.9) | 1,220(49.1) | |
| | Older adult (50–64) | 1,339(50.1) | 1,331(49.9) | |

1) *: p < 0.05, **: p < 0.01, ***: p < 0.001

### Associations of gender and age with influenza vaccination behavior

Table 3 shows results of analyzing the relationship between respondents' gender and age and influenza vaccination behavior (Table 3). There was a statistically significant association between age and influenza vaccination behavior in both men and women. The influenza vaccination rate was 14.2%p higher for older men than for younger men. It was 19.0%p higher for older women than for younger women. For both men and women, as age increased, more people received influenza vaccination.

### Behavioral determinants of influenza vaccination by age in men

Results of a binomial logistic regression analysis to identify behavioral determinants that might affect influenza vaccination status by age in men are shown in Table 4 (Table 4). The higher the level of education, the more likely a respondent to be vaccinated against influenza by 1.089 times (95% confidence interval [CI]: 1.000–1.185). Married people were 1.619 times more likely to receive influenza vaccination than single people (95% CI: 1.413–1.856). The group with chronic diseases was 1.248 times more likely to be vaccinated than the healthy group (95% CI: 1.093–1.425). The likelihood of smokers receiving influenza vaccination was 0.822 times lower than that of non-smokers (95% CI: 0.732–0.923). Binge drinkers were 0.749 times less likely to receive influenza vaccination than non-binge drinkers (95% CI: 0.650–0.864). The greater the number of COVID-19 vaccinations, the more likely a respondent to receive influenza vaccination by 1.881 times (95% CI: 1.725–2.052).

When respondents were stratified by age group, the following results were obtained. Among young men, married people were 1.480 times more likely to receive influenza vaccination than single men (95% CI: 1.131–1.935). The group that practiced physical activity was 1.381 times more likely to receive influenza vaccination than the group that did not (95% CI: 1.053–1.811). The greater the number of COVID-19 vaccinations, the more likely a respondent to receive influenza vaccination by 1.728 times (95% CI: 1.476–2.024). For middle-aged men, the higher the income level, the more likely a respondent to receive influenza vaccination by 1.144 times (95% CI: 1.002–1.306). Married people were 1.577 times more likely to receive influenza vaccination than single men (95% CI: 1.285–1.935). Smokers were 0.765 times less likely to receive influenza vaccination than non-smokers (95% CI: 0.624–0.937). Binge drinkers were 0.685 times less likely to receive influenza vaccination than non-binge drinkers (95% CI: 0.538–0.872). The greater the number of COVID-19 vaccinations, the more likely a respondent to receive influenza vaccination by 1.933 times (95% CI: 1.650–2.264). Among older men, married people were 1.521 times more likely to receive influenza vaccination than single men (95% CI: 1.151–2.008). The group with chronic diseases was 1.339 times more likely to receive influenza vaccination than the healthy group (95% CI: 1.126–1.592). Smokers were 0.761 times less likely to receive influenza vaccination than non-smokers (95% CI: 0.631–0.919). The group with binge drinking was 0.750 times less likely to receive influenza vaccination than the group without binge drinking (95% CI: 0.598–0.939). The greater the number of COVID-19 vaccinations, the more likely a respondent to receive influenza vaccination by 2.063 times (95% CI: 1.787–2.382).

**Table 4. Binomial logistic regression analysis of behavioral determinants of influenza vaccination by age (Men, n = 6964).**

| | Category | Total | | Young adult (19–34 yrs) | | Middle-aged (35–49 yrs) | | Older adult (50–64 yrs) | |
|---|---|---|---|---|---|---|---|---|---|
| | | OR | 95% CI | OR | 95% CI | OR | 95% CI | OR | 95% CI |
| Predisposing factors | Age | 0.943 | 0.865-1.027 | | | | | | |
| Enabling factors | Income | 1.015 | 0.940-1.095 | 0.969 | 0.844-1.113 | 1.144* | 1.002-1.306 | 0.962 | 0.844-1.095 |
| | Education | 1.089* | 1.000-1.185 | 0.954 | 0.804-1.131 | 1.158 | 0.989-1.355 | 1.046 | 0.916-1.196 |
| | Marital status (Ref.: single) | 1.619*** | 1.413-1.856 | 1.480** | 1.131-1.935 | 1.577*** | 1.285-1.935 | 1.521** | 1.151-2.008 |
| Need factors | Chronic disease (Ref.: none) | 1.248*** | 1.093-1.425 | 0.874 | 0.533-1.433 | 1.242 | 0.982-1.571 | 1.339*** | 1.126-1.592 |
| | Smoking (Ref.: no) | 0.822*** | 0.732-0.923 | 0.978 | 0.788-1.214 | 0.765* | 0.624-0.937 | 0.761** | 0.631-0.919 |
| | Binge drinking (Ref.: no) | 0.749*** | 0.650-0.864 | 0.833 | 0.623-1.112 | 0.685** | 0.538-0.872 | 0.750* | 0.598-0.939 |
| | Physical activity (Ref.: no) | 1.085 | 0.954-1.235 | 1.381* | 1.053-1.811 | 1.137 | 0.916-1.412 | 0.928 | 0.754-1.143 |
| | COVID-19 vaccination | 1.881*** | 1.725-2.052 | 1.728*** | 1.476-2.024 | 1.933*** | 1.650-2.264 | 2.063*** | 1.787-2.382 |
| -2LL | | 8592.580 | | 2634.120 | | 2861.721 | | 3053.732 | |
| Nagelkerke R² | | 0.087 | | 0.042 | | 0.087 | | 0.091 | |

[1])*p<0.05, **p<0.01, ***p<0.001

[2])OR: Odds ratio

[3])95% CI: 95% confidence interval

### Behavioral determinants of influenza vaccination by age in women

Results of analyzing behavioral determinants affecting influenza vaccination by age of women respondents are shown in Table 5 (Table 5). The higher the level of education, the more likely a respondent to receive influenza vaccination by 1.168 times (95% CI: 1.075–1.270). Married women were 2.242 times more likely to receive influenza vaccination than single women (95% CI: 1.965–2.557). The group with chronic diseases was 1.282 times more likely to be vaccinated than the healthy group (95% CI: 1.101–1.494). The greater the number of COVID-19 vaccinations, the more likely a respondent to receive influenza vaccination by 1.852 times (95% CI: 1.712–2.003).

When stratified by age group, results are as follows. For young women, the higher their level of education, the more likely to receive influenza vaccination by 1.188 times (95% CI: 1.000–1.411). The likelihood of married women receiving influenza vaccination was 2.419 times higher than that of single women (95% CI: 1.942–3.015). Smokers were 0.551 times less likely to receive influenza vaccination than non-smokers (95% CI: 0.359–0.847). The greater the number of COVID-19 vaccinations, the more likely to receive influenza vaccination by 1.540 times (95% CI: 1.321–1.795). For middle-aged women, the higher their education level, the more likely to receive influenza vaccination by 1.190 times (95% CI: 1.029–1.376). Married women were 1.931 times more likely to receive influenza vaccination than single women (95% CI: 1.580–2.361). The greater the number of COVID-19 vaccinations, the more likely to receive influenza vaccination by 1.658 times (95% CI: 1.453–1.893). Among older women, the group with chronic diseases was 1.354 times (95% CI: 1.133–1.619) more likely to be vaccinated than the healthy group. The greater the number of COVID-19 vaccinations, the more likely to receive influenza vaccination by 2.557 times (95% CI: 2.229–2.934).

### Behavioral determinants of influenza vaccination by age with interaction terms between gender and need/enabling factors

The results of the interaction effects between gender and enabling/need factors by age group are presented in Table 6 (Table 6). Overall, the trends observed in the full sample, regardless of gender, were consistent with those reported in Tables 4 and 5. Specifically, among the enabling factors, respondents with higher education levels were more likely to have received SIV (odds ratio [OR]=1.089; 95% CI: 1.000–1.185). Additionally, married individuals were more likely to be vaccinated than their single counterparts (OR=1.615; 95% CI: 1.426–1.829). Among the need factors, respondents with

**Table 5. Binomial logistic regression analysis of behavioral determinants of influenza vaccination by age (Women, n = 7571).**

| | Category | Total | | Young adult (19–34 yrs) | | Middle-aged (35–49 yrs) | | Older adult (50–64 yrs) | |
|---|---|---|---|---|---|---|---|---|---|
| | | OR | 95% CI | OR | 95% CI | OR | 95% CI | OR | 95% CI |
| Predisposing factor | Age | 0.948 | 0.874-1.040 | | | | | | |
| Enabling factors | Income | 0.968 | 0.902-1.040 | 0.999 | 0.874-1.142 | 1.032 | 0.913-1.166 | 0.978 | 0.868-1.102 |
| | Education | 1.168*** | 1.075-1.270 | 1.188* | 1.000-1.411 | 1.190* | 1.029-1.376 | 0.918 | 0.798-1.056 |
| | Marital status (Ref.: single) | 2.242*** | 1.965-2.557 | 2.419*** | 1.942-3.015 | 1.931*** | 1.580-2.361 | 1.044 | 0.752-1.449 |
| Need factors | Chronic disease (Ref.: none) | 1.282*** | 1.101-1.494 | 1.137 | 0.495-2.611 | 1.282 | 0.903-1.821 | 1.354*** | 1.133-1.619 |
| | Smoking (Ref.: no) | 0.775 | 0.586-1.026 | 0.551** | 0.359-0.847 | 0.893 | 0.517-1.541 | 1.295 | 0.711-2.360 |
| | Binge drinking (Ref.: no) | 0.931 | 0.747-1.161 | 1.089 | 0.792-1.496 | 0.698 | 0.479-1.018 | 1.113 | 0.645-1.920 |
| | Physical activity (Ref.: no) | 1.038 | 0.930-1.160 | 0.974 | 0.788-1.205 | 1.148 | 0.958-1.377 | 1.049 | 0.868-1.268 |
| | COVID-19 vaccination | 1.852*** | 1.712-2.003 | 1.540*** | 1.321-1.795 | 1.658*** | 1.453-1.893 | 2.557*** | 2.229-2.934 |
| -2LL | | 9785.298 | | 2881.125 | | 3323.085 | | 3458.264 | |
| Nagelkerke R² | | 0.100 | | 0.061 | | 0.062 | | 0.116 | |

1)*p < 0.05, **p < 0.01, ***p < 0.001

2)OR: Odds ratio

3)95% CI: 95% confidence interval

chronic diseases had a higher likelihood of receiving SIV compared to those without chronic conditions (OR=1.247; 95% CI: 1.095–1.420). However, respondents who smoked (OR=0.822; 95% CI: 0.732–0.923) or engaged in binge drinking (OR=0.749; 95% CI: 0.650–0.864) were less likely to receive SIV than non-smokers and non-binge drinkers. Furthermore, individuals who had been vaccinated against COVID-19 were significantly more likely to have received SIV compared to those who had not (OR=1.880; 95% CI: 1.726–2.048).

Regarding the interaction effect between gender and these factors, young married women were 1.635 times more likely to be vaccinated with SIV than the rest of the same age group (95% CI: 1.156–2.314). Young female smokers were significantly less likely to receive SIV than male non-smokers of the same age (OR = 0.564; 95% CI: 0.349–0.912). Among older adults, women who had received the COVID-19 vaccine were 1.240 times more likely to have been vaccinated against seasonal influenza than their male counterparts who had not received the COVID-19 vaccine (95% CI: 1.016–1.512).

## Discussion

SIV is an important practice to prevent complications or death due to influenza virus infection and to reduce the burden of medical expenses [30]. South Korea has also made efforts to minimize the burden of disease in high-risk groups through a national vaccination program. However, the influenza vaccination rate among adults was less than 30% [31] despite a systematic surveillance system and excellent accessibility to medical services. Therefore, this study applied Andersen's model to identify contextual determinants that might affect SIV and sought ways to increase vaccination rates.

Major findings of this study are as follows. In terms of enabling factors, the higher the level of education for both men and women respondents, the more likely they were to receive SIV. Compared to singles, couples of men and women respondents were more likely to be vaccinated. In terms of need factors, for both men and women respondents, groups with chronic diseases were more likely to receive influenza vaccination than groups without chronic diseases. Additionally, the greater the number of COVID-19 vaccinations, the higher the likelihood of receiving influenza vaccination. Nonetheless, smoking and drinking were significant variables only for men. Men who smoked or drank alcohol heavily were less likely to be vaccinated against influenza than men who did not smoke or drank alcohol.

**Table 6. Binomial logistic regression analysis of behavioral determinants of seasonal influenza vaccination with interaction terms between gender and need/enabling factors (n = 14,535).**

| Category | | Total | | Young adult (19–34 yrs) | | Middle-aged (35–49 yrs) | | Older adult (50–64 yrs) | |
|---|---|---|---|---|---|---|---|---|---|
| | | OR | 95% CI | OR | 95% CI | OR | 95% CI | OR | 95% CI |
| Predisposing factors | Age | 0.945 | 0.891-1.003 | | | | | | |
| | Gender (Ref.: men) | 1.094 | 0.707-1.692 | 1.192 | 0.535-2.655 | 2.150 | 0.987-4.685 | 1.045 | 0.450-2.430 |
| Enabling factors | Income | 1.015 | 0.941-1.095 | 0.969 | 0.844-1.113 | 1.144* | 1.002-1.306 | 0.962 | 0.844-1.095 |
| | Education | 1.089* | 1.000-1.185 | 0.954 | 0.804-1.131 | 1.158 | 0.989-1.355 | 1.046 | 0.916-1.196 |
| | Marital status (Ref.: single) | 1.615*** | 1.426-1.829 | 1.480** | 1.131-1.935 | 1.577*** | 1.285-1.935 | 1.521** | 1.151-2.008 |
| Need factors | Chronic disease (Ref.: none) | 1.247*** | 1.095-1.420 | 0.874 | 0.533-1.433 | 1.242 | 0.982-1.571 | 1.339*** | 1.126-1.592 |
| | Smoking (Ref.: no) | 0.822*** | 0.732-0.923 | 0.978 | 0.788-1.214 | 0.765** | 0.624-0.937 | 0.761** | 0.631-0.919 |
| | Binge drinking (Ref.: no) | 0.749*** | 0.650-0.864 | 0.833 | 0.623-1.112 | 0.685** | 0.538-0.872 | 0.750* | 0.598-0.939 |
| | Physical activity (Ref.: no) | 1.086 | 0.954-1.236 | 1.381* | 1.053-1.811 | 1.137 | 0.916-1.412 | 0.928 | 0.754-1.143 |
| | COVID-19 vaccination | 1.880*** | 1.726-2.048 | 1.728*** | 1.476-2.024 | 1.933*** | 1.650-2.264 | 2.063*** | 1.787-2.382 |
| Gender X Income | | 0.954 | 0.860-1.058 | 1.030 | 0.850-1.249 | 0.902 | 0.753-1.081 | 1.017 | 0.852-1.213 |
| Gender X Education | | 1.072 | 0.953-1.207 | 1.245 | 0.977-1.588 | 1.028 | 0.829-1.273 | 0.877 | 0.723-1.065 |
| Gender X Marital status | | 1.392*** | 1.193-1.623 | 1.635** | 1.156-2.314 | 1.225 | 0.919-1.632 | 0.686 | 0.446-1.055 |
| Gender X Chronic disease | | 1.030 | 0.847-1.252 | 1.301 | 0.494-3.423 | 1.033 | 0.677-1.575 | 1.012 | 0.789-1.297 |
| Gender X Smoking | | 0.943 | 0.696-1.277 | 0.564* | 0.349-0.912 | 1.168 | 0.652-2.091 | 1.701 | 0.907-3.189 |
| Gender X Binge drinking | | 1.242 | 0.955-1.614 | 1.308 | 0.851-2.010 | 1.020 | 0.652-1.595 | 1.485 | 0.823-2.678 |
| Gender X Physical activity | | 0.957 | 0.807-1.134 | 0.706* | 0.500-0.996 | 1.010 | 0.761-1.340 | 1.130 | 0.853-1.498 |
| Gender X COVID-19 vaccination | | 0.985 | 0.880-1.104 | 0.891 | 0.715-1.110 | 0.858 | 0.698-1.054 | 1.240* | 1.016-1.512 |
| -2LL | | 18377.886 | | 5515.245 | | 6184.805 | | 6511.996 | |
| Nagelkerke R² | | 0.102 | | 0.054 | | 0.089 | | 0.113 | |

[1)]*p < 0.05, **p < 0.01, ***p < 0.001

[2)]OR: Odds ratio

[3)]95% CI: 95% confidence interval

Previous studies have reported that predisposing factors such as race, age, and gender can affect SIV [10,32,33]. Conversely, in terms of enabling factors, the association between socioeconomic status and SIV differed by country [10,25,26,34]. On the other hand, need factors are more important for SIV than predisposing or enabling factors [20,35]. Vaccination rates tend to increase as individuals perceive a high risk of disease infection or use more medical care [32,36]. Despite being classified as a high-risk group for seasonal influenza infection, people with chronic diseases have a low vaccination rate, even with regular access to medical care [37]. Health behaviors such as smoking, drinking, and physical activity are not only associated with influenza vaccination rate, but also highly correlated with chronic diseases. They are important factors for explaining the level of primary health care utilization [25,38,39]. Among young cancer survivors in Korea, the influenza vaccination rate was lower among those who were smokers and those who had been diagnosed with cancer for more than 10 years. It was also lower in the middle-aged group than in the elderly with disabilities [40,41]. To increase seasonal influenza vaccination rate, it is necessary to use an integrated behavioral model to identify characteristics of the target population and implement a customized campaign. When respondents were stratified by gender and age as predisposing factors, results are as follows. In terms of enabling factors, married men were consistently more likely to receive influenza vaccination than single men across all age groups. In terms of need factors, men were more likely to receive influenza vaccination as the number of COVID-19 vaccinations increased across all age groups. Meanwhile, middle-aged people who smoked or drank alcohol heavily were less likely to receive influenza vaccination

than those who did not. On the other hand, young people who engaged in physical activity were more likely to receive influenza vaccination than those who did not. Nevertheless, the factors influencing influenza vaccination among women were relatively more complex than those for men. In terms of need factors, the greater the number of COVID-19 vaccinations in all age groups, the higher the likelihood of receiving influenza vaccination, which was the same as for men. For women, however, the behavioral determinants of influenza vaccination differed between those aged 50 years or older and those under 50. Women under the age of 50 were more likely to be vaccinated against influenza if they were married or had a higher level of education. On the other hand, women in their 50s or older were more likely to get vaccinated if they had a chronic disease, whereas their educational attainment and marital status had no effect on vaccination rate. In other words, enabling factors were important for women under 50 years of age. By contrast, need factors played a more significant role for women aged 50 years or older.

The empirical findings of this study indicate that higher education, marital status, chronic diseases, and a greater number of COVID-19 vaccinations significantly increase the likelihood of influenza vaccination, whereas smoking and binge drinking reduce it. Age-stratified analyses further revealed that physical activity and income level had a greater influence on younger and middle-aged men, while education and chronic diseases were more significant factors for younger and older women. Notably, across all age groups, the number of COVID-19 vaccinations consistently increased the likelihood of influenza vaccination. These findings underscore the need for gender-specific vaccination campaigns. For men, vaccine hesitancy among smokers and binge drinkers can be addressed through behavioral health interventions. For women, vaccination messaging can be reinforced through healthcare provider recommendations and chronic disease management programs.

This study suggests several practical implications in increasing VCR. First, since higher education correlates with increased influenza vaccination rates in both genders, enhancing health literacy through targeted campaigns is essential. Second, as the vaccination rate among singles is lower than that in married couples for both men and women, there is a need to inform them of the importance of influenza vaccination. Third, since a respondent's income level was not a significant factor in determining influenza vaccination, non-price policies to increase the vaccination rate should be considered important. Fourth, elderly people with chronic diseases have a high vaccination rate for both men and women. Clinically, they are a vulnerable group who can be hospitalized with severe illness if infected with influenza. Thus, we need to look further into factors that can increase their vaccination rate. Fifth, among middle-aged men, those who smoke or drink alcohol heavily had significantly lower vaccination rates. They are likely to be a group clustered with various health risk behaviors [42]. Therefore, it is necessary to manage these people as a high-risk group in conjunction with health promotion projects and encourage influenza vaccination. Sixth, in the case of young people, men who exercised and women who smoked were groups that differed from each other in terms of influenza vaccination. Therefore, for the younger generation, vaccination rates can be improved through targeted strategies aligned with their preferences. Seventh, COVID-19 vaccination practice was closely related to influenza vaccination across all ages and genders. This shows that the act of vaccination can ultimately be promoted or hindered by various factors surrounding the act, not the type of vaccine. Therefore, like other health care services, it is ultimately necessary to improve various health care environments that can increase vaccination rates.

This study has several limitations. First, effects of health policies and social systems that can affect vaccination could not be reflected in the statistical model of this study. Therefore, multilevel studies are needed to understand decision-making related to vaccination. Second, awareness or attitudes toward SIV might have changed due to COVID-19. Therefore, longitudinal research about the impact of infectious disease pandemics on vaccination is needed. Third, medical use behaviors such as vaccinations are influenced by social and cultural factors. Therefore, cross-national comparative research is needed on anti-vaccine activism and the impact of fake news and ghost stories about side effects.

Despite many countries operating national vaccination programs targeting the entire population, vaccination rates are still not high [34]. Due to COVID-19, South Korea provided free seasonal influenza vaccination to adults aged 62–64 and

children aged 13–18 in the 2020–2021 season. Yet, the vaccination rate did not exceed that of previous years [9]. This suggests that additional interventions other than price are needed to improve vaccination rates. This study empirically showed that influenza vaccination was affected by predisposing, enabling, and need factors. To effectively intervene in individual health behaviors, it is necessary to identify characteristics of the population group, use segmented strategies, and provide customized messages [43].

## Supporting information

**S1 File. Raw research data with identifying information removed.**
(CSV)

## Author contributions

**Conceptualization:** Minsoo Jung.

**Data curation:** So-Hyun Kim.

**Formal analysis:** So-Hyun Kim.

**Methodology:** So-Hyun Kim, Minsoo Jung.

**Software:** So-Hyun Kim.

**Supervision:** Minsoo Jung.

**Validation:** Minsoo Jung.

**Visualization:** So-Hyun Kim.

**Writing – original draft:** So-Hyun Kim, Minsoo Jung.

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
