## [Decision Letter · Decision Letter 0]

30 Jan 2025

PONE-D-24-58260Disentangling behavioral determinants of seasonal influenza vaccination in post-corona era: An integrated model approachPLOS ONE

Dear Dr. Jung,

Thank you for submitting your manuscript to PLOS ONE. After careful consideration, we feel that it has merit but does not fully meet PLOS ONE’s publication criteria as it currently stands. Therefore, we invite you to submit a revised version of the manuscript that addresses the points raised during the review process.

We look forward to receiving your revised manuscript.

Kind regards,

Hani Amir Aouissi

Academic Editor

PLOS ONE

Journal Requirements:

Reviewers' comments:

Reviewer's Responses to Questions

**Comments to the Author**

1. Is the manuscript technically sound, and do the data support the conclusions?

Reviewer #1: Yes

Reviewer #2: Yes

2. Has the statistical analysis been performed appropriately and rigorously? 

Reviewer #1: Yes

Reviewer #2: Yes

3. Have the authors made all data underlying the findings in their manuscript fully available?

Reviewer #1: Yes

Reviewer #2: Yes

4. Is the manuscript presented in an intelligible fashion and written in standard English?

Reviewer #1: Yes

Reviewer #2: Yes

5. Review Comments to the Author

Reviewer #1: Dear editors and authors,

Thank you for the opportunity to review the manuscript « Disentangling behavioral determinants of seasonal influenza vaccination in post-corona era: An integrated model approach », with the reference « PONE-D-24-58260 ».

The article was pleasant to read, the study was complete and well handled, I think it is suitable for publication after some correction.

Consequently, I just have some comments, I hope it will help the authors to improve the overall level of the manuscript.

* Please write the full term « Coronavirus disease » before the acronym COVID-19, as it was the first time it appears in the text.

* Please consider including more results in the abstract.

* I think you need an additional reference in the first part of your introduction, as this background should be referenced.

* It may be clearer for the readers if the objective was clearly defined in the last paragraph of the introduction section

* Why more than 19 years old ? is it something specific to Korea ?

* Please put a little explanation under the title measures explaining that you will explain dependent and independant variables

* In the discussion section (it is also the same case for the introduction section), you used a lot the term « however », please try to replace it in some parts by synonyms (eg. Nevertheless, nonetheless…etc.).

* I think the first line of tables should be put in bold, please try to refer to the instructions to authors in website of the journal.

Sincerely.

Reviewer #2: Dear Authors,

Thank you for your submission of the manuscript titled "Disentangling behavioral determinants of seasonal influenza vaccination in post-corona era: An integrated model approach." Your study presents a valuable and timely analysis of the factors influencing seasonal influenza vaccination, particularly in the evolving public health landscape shaped by the COVID-19 pandemic.

The use of Andersen’s behavioral model provides a strong theoretical foundation, effectively categorizing predisposing, enabling, and need-based factors influencing vaccination decisions. Additionally, the utilization of a large, nationally representative dataset (n = 14,535) from the 2022 Community Health Survey strengthens the credibility of the findings. The application of binomial logistic regression analysis, along with stratification by age and gender, further enhances the robustness of the study.

The manuscript contributes important public health insights, particularly in identifying key determinants such as education, income, marital status, chronic disease presence, smoking, and COVID-19 vaccination history. These findings offer actionable implications for policymakers aiming to increase influenza vaccination coverage.

However, there are some areas where the manuscript could be improved to further enhance clarity, depth of analysis, and practical impact. In particular, more explicit gender-specific data presentation, statistical comparisons, and discussion on behavioral risks would strengthen the manuscript. Below are our recommendations for refinement.

1. Enhanced Comparability: Presenting Gender-Specific Distributions

• Current Issue: Table 1 & 2 presents aggregate percentages for key variables without distinguishing among gender variation.

• Recommendation:

o Split Table 1 by gender to show separate distributions for men and women.

o This will allow a clearer comparison of how different factors influence influenza vaccination uptake among men and women.

2. Improved Statistical Analysis and Interpretation

• Current Issue: The manuscript does not explicitly analyze gender-specific trends separately in the statistical models.

• Recommendation:

o Perform separate logistic regression models for male and female to determine whether behavioral determinants impact them differently.

o Alternatively, include interaction terms (e.g., gender × smoking, gender × chronic disease) in the analysis to assess whether gender moderates the effect of these factors.

3. Strengthening the Discussion on Behavioral Risks and Correlations

• Current Issue: While the results mention some gender differences, the discussion does not fully explore how factors like smoking, alcohol consumption, and chronic disease presence impact male and female differently.

• Recommendation:

o Explicitly discuss how smoking and binge drinking may reduce vaccination uptake more significantly in males due to lower health-seeking behavior.

o Highlight that females with chronic diseases may be more likely to get vaccinated due to increased health awareness and regular medical visits.

o Provide public health recommendations for gender-specific vaccination strategies, such as targeted outreach campaigns for males engaging in risk behaviors.

4. Public Health Interventions: Gender-Specific Vaccination Campaigns

• Current Issue: The manuscript does not provide actionable gender-based strategies for improving vaccination rates.

• Recommendation:

o Discuss the need for gender-tailored public health campaigns, such as:

For males: Address vaccine hesitancy among smokers and binge drinkers through behavioral health interventions.

For females: Strengthen vaccination messaging through healthcare provider recommendations and chronic disease management programs.

5. Language and Clarity:

• While the manuscript is generally well-written, some sentences are verbose and can be rephrased for conciseness. (See specific language corrections below with line numbers.)

• Ensure consistency in terminology, particularly in descriptions of "enabling," "predisposing," and "need factors."

6. Integration of Findings:

• While the results section is detailed, the discussion could better integrate findings into broader public health implications, especially regarding the role of health literacy and cultural attitudes in vaccination behavior.

7. Data Representation:

• Ensure all figures and tables are formatted consistently. Some legends and titles can be shortened for clarity.

Incorporating these recommendations will strengthen the manuscript’s contribution by offering a clearer understanding of how gender influences influenza vaccination behavior, along with the flowing suggestions for improvement in language

Abstract

• Page-3; Line 3:

"In the post-COVID-19 era, the importance of vaccination and group immunity has grown."

• Page-3; Line 13:

"Binomial logistic regression revealed that higher education, marital status, chronic diseases, and more COVID-19 vaccinations significantly increased the likelihood of influenza vaccination, while smoking and binge drinking reduced it."

Introduction

• Page-5; Line 2:

"Influenza remains a major public health threat despite the availability of vaccines and antiviral medications, causing significant morbidity and mortality, especially among vulnerable populations."

• Page-7; Line 7:

"Vaccination involves time, costs, and potential side effects, and individuals may still contract influenza despite being vaccinated."

Methods

• Page-9; Line 2:

"Consequently, the Community Health Survey used a multistage, stratified, random sampling method to represent the broader Korean population."

Discussion

• Page-22; Line 19:

"Since higher education correlates with increased influenza vaccination rates in both genders, enhancing health literacy through targeted campaigns is essential."

• Page-13; Line 11:

"For the younger generation, vaccination rates can be improved through targeted strategies aligned with their preferences."

6. PLOS authors have the option to publish the peer review history of their article (what does this mean? ). If published, this will include your full peer review and any attached files.

**Do you want your identity to be public for this peer review?** For information about this choice, including consent withdrawal, please see our Privacy Policy .

Reviewer #1: No

Reviewer #2: **Yes: ** Muhammad Athar Abbas (DVM, PhD)

---

## [Author Response · Author response to Decision Letter 1]

27 Feb 2025

Point-by-point responses to reviewers’ comments

Ref: PONE-D-24-58260

Disentangling behavioral determinants of seasonal influenza vaccination in post-corona era: An integrated model approach

Dear Dr. Hani Amir Aouissi, Academic Editor of PLOS ONE

Thank you for your letter dated January 31, 2025. We have thoroughly addressed the points raised by the reviewers and supplemented the manuscript accordingly. We believe that the manuscript now meets the publication criteria of PLOS ONE. Below are the details of how we responded to each of the reviewers' comments. Given the extended wait for these results, we kindly request an expedited review of this revision. Thank you for your time and consideration.

Very best,

Authors

Reviewer #1

Thank you for the opportunity to review the manuscript « Disentangling behavioral determinants of seasonal influenza vaccination in post-corona era: An integrated model approach », with the reference « PONE-D-24-58260 ». The article was pleasant to read, the study was complete and well handled, I think it is suitable for publication after some correction. Consequently, I just have some comments, I hope it will help the authors to improve the overall level of the manuscript.

>>> Authors’ Response: Thank you for your thoughtful feedback and encouragement. We have revised and expanded the paper in response to your detailed comments, as outlined below.

1. Please write the full term « Coronavirus disease » before the acronym COVID-19, as it was the first time it appears in the text.

>>> Authors’ Response: As the reviewer noted, we have provided the full term upon the first occurrence of each abbreviation.

2. Please consider including more results in the abstract.

>>> Authors’ Response: We have revised the Results section of the Abstract to incorporate key figures from the study’s findings. These updates ensure a clearer presentation of the major results and enhance the paper’s originality.

3. I think you need an additional reference in the first part of your introduction, as this background should be referenced.

>>> Authors’ Response: We have added four references to strengthen the argument in the introduction and enrich the background of this study, thereby enhancing readers' understanding.

4. It may be clearer for the readers if the objective was clearly defined in the last paragraph of the introduction section.

>>> Authors’ Response: We have revised and expanded the final paragraph of the Introduction in the revised manuscript. The updated paragraph explicitly articulates the study's purpose, enhancing clarity and readability for readers.

5. Why more than 19 years old? is it something specific to Korea?

>>> Authors’ Response: In South Korea, individuals under the age of 18 are legally considered minors, resulting in distinct sample characteristics compared to adults. They are overseen separately by the Korean Immunization Committee, and parental consent is required for participation in surveys.

6. Please put a little explanation under the title measures explaining that you will explain dependent and independent variables.

>>> Authors’ Response: As per the reviewer's recommendation, we have incorporated this explanation into the revised manuscript.

7. In the discussion section (it is also the same case for the introduction section), you used a lot the term « however », please try to replace it in some parts by synonyms (eg. Nevertheless, nonetheless…etc.).

>>> Authors’ Response: Thank you for your careful comments. We have replaced eight instances of 'However' in the Discussion section with more appropriate terms or restructured sentences for improved clarity. These revisions have enhanced the overall flow and readability of the manuscript.

8. I think the first line of tables should be put in bold, please try to refer to the instructions to authors in website of the journal.

>>> Authors’ Response: We formatted the first row of the table in bold to align with the journal's author guidelines.

Reviewer #2

Thank you for your submission of the manuscript titled "Disentangling behavioral determinants of seasonal influenza vaccination in post-corona era: An integrated model approach." Your study presents a valuable and timely analysis of the factors influencing seasonal influenza vaccination, particularly in the evolving public health landscape shaped by the COVID-19 pandemic. The use of Andersen’s behavioral model provides a strong theoretical foundation, effectively categorizing predisposing, enabling, and need-based factors influencing vaccination decisions. Additionally, the utilization of a large, nationally representative dataset (n = 14,535) from the 2022 Community Health Survey strengthens the credibility of the findings. The application of binomial logistic regression analysis, along with stratification by age and gender, further enhances the robustness of the study. The manuscript contributes important public health insights, particularly in identifying key determinants such as education, income, marital status, chronic disease presence, smoking, and COVID-19 vaccination history. These findings offer actionable implications for policymakers aiming to increase influenza vaccination coverage. However, there are some areas where the manuscript could be improved to further enhance clarity, depth of analysis, and practical impact. In particular, more explicit gender-specific data presentation, statistical comparisons, and discussion on behavioral risks would strengthen the manuscript. Below are our recommendations for refinement.

>>> Authors’ Response: Thank you for your positive assessment of the importance and value of this paper. We believe that this study provides essential insights for preparing for the post-COVID-19 era. In response to the reviewer's comments, we have carefully revised and improved the manuscript as outlined below.

1. Enhanced Comparability: Presenting Gender-Specific Distributions

• Current Issue: Table 1 & 2 presents aggregate percentages for key variables without distinguishing among gender variation.

• Recommendation:

o Split Table 1 by gender to show separate distributions for men and women.

o This will allow a clearer comparison of how different factors influence influenza vaccination uptake among men and women.

>>> Authors’ Response: As recommended by the reviewer, we have added a table that breaks down gender differences. However, stratifying the table by gender removes gender from the list of predisposing factors. Therefore, we have presented the gender-stratified table as Appendix 1. Similarly, the vaccination status results stratified by gender from Table 2 are presented in Appendix 2. We agree with the reviewer's point; however, since gender is also a predisposing factor, we have presented both table formats to offer readers a more comprehensive view of the data.

2. Improved Statistical Analysis and Interpretation

• Current Issue: The manuscript does not explicitly analyze gender-specific trends separately in the statistical models.

• Recommendation:

o Perform separate logistic regression models for male and female to determine whether behavioral determinants impact them differently.

o Alternatively, include interaction terms (e.g., gender × smoking, gender × chronic disease) in the analysis to assess whether gender moderates the effect of these factors.

>>> Authors’ Response: We have already presented logistic regression models stratified by sex to identify gender-specific traits. However, the reviewers suggested further analysis of how behavioral determinants influence vaccination by incorporating interaction terms in the model. In response, we added Table 6 to examine how sex moderates enabling and needs factors.

3. Strengthening the Discussion on Behavioral Risks and Correlations

• Current Issue: While the results mention some gender differences, the discussion does not fully explore how factors like smoking, alcohol consumption, and chronic disease presence impact male and female differently.

• Recommendation:

o Explicitly discuss how smoking and binge drinking may reduce vaccination uptake more significantly in males due to lower health-seeking behavior.

o Highlight that females with chronic diseases may be more likely to get vaccinated due to increased health awareness and regular medical visits.

o Provide public health recommendations for gender-specific vaccination strategies, such as targeted outreach campaigns for males engaging in risk behaviors.

>>> Authors’ Response: Based on the results of Analysis 2, we incorporated a literature review on behavioral risks and correlations in the discussion section. For men, we discussed the impact of smoking and binge drinking on vaccination rates, while for women, we examined whether having a chronic disease increases the likelihood of vaccination. Additionally, we proposed tailored vaccination strategies based on gender and age.

4. Public Health Interventions: Gender-Specific Vaccination Campaigns

• Current Issue: The manuscript does not provide actionable gender-based strategies for improving vaccination rates.

• Recommendation:

o Discuss the need for gender-tailored public health campaigns, such as:

For males: Address vaccine hesitancy among smokers and binge drinkers through behavioral health interventions.

For females: Strengthen vaccination messaging through healthcare provider recommendations and chronic disease management programs.

>>> Authors’ Response: In response to the reviewers' comments, we have revised the manuscript to include actionable gender-based strategies for improving vaccination coverage. For men, we address vaccine hesitancy through behavioral health interventions, such as smoking cessation and moderate alcohol consumption. For women, we emphasize reinforcing vaccination messages within chronic disease management programs.

5. Language and Clarity:

• While the manuscript is generally well-written, some sentences are verbose and can be rephrased for conciseness. (See specific language corrections below with line numbers.)

• Ensure consistency in terminology, particularly in descriptions of "enabling," "predisposing," and "need factors."

>>> Authors’ Response: We thoroughly reread the manuscript, refined the wording, streamlined sentences, and ensured consistency in terminology and conceptual clarity.

6. Integration of Findings:

• While the results section is detailed, the discussion could better integrate findings into broader public health implications, especially regarding the role of health literacy and cultural attitudes in vaccination behavior.

>>> Authors’ Response: In the Discussion section, we have expanded the content to link our findings to broader public health implications. We aimed to highlight this study's contribution to understanding the role of health literacy and cultural attitudes in vaccination behavior.

7. Data Representation:

• Ensure all figures and tables are formatted consistently. Some legends and titles can be shortened for clarity.

>>> Authors’ Response: We thoroughly reviewed all figures and tables to ensure compliance with the journal's standards, shortening titles and legends where necessary.

Incorporating these recommendations will strengthen the manuscript’s contribution by offering a clearer understanding of how gender influences influenza vaccination behavior, along with the flowing suggestions for improvement in language.

>>> Authors’ Response: We have carefully reviewed all the reviewers' recommendations and incorporated the necessary revisions into the manuscript. Additionally, we have adopted the refined expressions you suggested to enhance the contextual fit of the article. Gender is a key predisposing factor in our analytical framework. However, to ensure a more comprehensive perspective, we have also considered age, another predisposing factor, in our examination of the determinants of influenza vaccination. Therefore, our findings indicated that the enabling and need factors outlined in the Andersen Model exert different effects depending on an individual's gender and age. We sincerely appreciate your thorough review and the valuable, constructive feedback you have provided.

---

## [Decision Letter · Decision Letter 1]

4 Apr 2025

Disentangling behavioral determinants of seasonal influenza vaccination in post-corona era: An integrated model approach

PONE-D-24-58260R1

Dear Dr. Jung,

We’re pleased to inform you that your manuscript has been judged scientifically suitable for publication and will be formally accepted for publication once it meets all outstanding technical requirements.

Kind regards,

Hani Amir Aouissi

Academic Editor

PLOS ONE

Additional Editor Comments (optional):

Reviewers' comments:

Reviewer's Responses to Questions

**Comments to the Author**

1. If the authors have adequately addressed your comments raised in a previous round of review and you feel that this manuscript is now acceptable for publication, you may indicate that here to bypass the “Comments to the Author” section, enter your conflict of interest statement in the “Confidential to Editor” section, and submit your "Accept" recommendation.

Reviewer #1: All comments have been addressed

Reviewer #3: (No Response)

2. Is the manuscript technically sound, and do the data support the conclusions?

Reviewer #1: Yes

Reviewer #3: Yes

3. Has the statistical analysis been performed appropriately and rigorously? 

Reviewer #1: Yes

Reviewer #3: Yes

4. Have the authors made all data underlying the findings in their manuscript fully available?

Reviewer #1: Yes

Reviewer #3: (No Response)

5. Is the manuscript presented in an intelligible fashion and written in standard English?

Reviewer #1: Yes

Reviewer #3: Yes

6. Review Comments to the Author

Reviewer #1: I read the revised version « PONE-D-24-58260R1 » of the manuscript entitled « Disentangling behavioral

determinants of seasonal influenza vaccination in post-corona era: An integrated model approach »

submitted for publication in PLOS ONE journal.

The authors made an effort represented by significant corrections in their manuscript specifically in the

results and discussion sections, while taking into consideration my suggestions and comments, in

addition to those of other reviewers. The responses to reviewers’ comments were also adequate, polite

and well structured.

I have no more comments. I think the overall level of the paper is quite high and merit publication in this

valuable journal. Sincerely.

Reviewer #3: I just read the article referenced with the code « PONE-D-24-58260R1 » about SIV in South Korea.

Apparently, it is a second revised version, and I saw in the R1 version that authors responded to the comments of previous reviewers, gave additionnal explanations, and made a lot of corrections.

According to what I understand, the study demonstrated that SIV in South Korea is influenced by behavior, socio-demographic factors and COVID-related factors such as vaccination. It was interesting to notice the variations by status, age, gender…etc.

I also agree with the authors point of view recommeding increasing public health awereness and applying strategies and free programs to improve vaccination rates.

The statiscital aspect was well handled, I have no comment. I can only recommend the paper because I think it is very interesting.

I just don’t know why the sentence « in the post-COVID-19 era, the importance of vaccination and group immunity has grown » in the abstract is in italics. I recommend to fix it before publication if possible.

7. PLOS authors have the option to publish the peer review history of their article (what does this mean? ). If published, this will include your full peer review and any attached files.

**Do you want your identity to be public for this peer review?** For information about this choice, including consent withdrawal, please see our Privacy Policy .

Reviewer #1: No

Reviewer #3: No

---

## [Editor Report · Acceptance letter]

PONE-D-24-58260R1

PLOS ONE

Dear Dr. Jung,

I'm pleased to inform you that your manuscript has been deemed suitable for publication in PLOS ONE. Congratulations! Your manuscript is now being handed over to our production team.

Kind regards,

on behalf of

Dr. Hani Amir Aouissi

Academic Editor

PLOS ONE